# Effect of a Simulated Heat Wave on Physiological Strain and Labour Productivity

**DOI:** 10.3390/ijerph18063011

**Published:** 2021-03-15

**Authors:** Leonidas G. Ioannou, Konstantinos Mantzios, Lydia Tsoutsoubi, Zoe Panagiotaki, Areti K. Kapnia, Ursa Ciuha, Lars Nybo, Andreas D. Flouris, Igor B. Mekjavic

**Affiliations:** 1Department of Automation, Biocybernetics and Robotics, Jozef Stefan Institute, 1000 Ljubljana, Slovenia; ursa.ciuha@gmail.com (U.C.); igor.mekjavic@ijs.si (I.B.M.); 2FAME Laboratory, Department of Physical Education and Sport Science, University of Thessaly, 42131 Trikala, Greece; konstantinosmantzios@gmail.com (K.M.); lydiatsoutsoubi@gmail.com (L.T.); zwhpanagiot@gmail.com (Z.P.); Areti.kapnia@gmail.com (A.K.K.); andreasflouris@gmail.com (A.D.F.); 3Department of Nutrition, Exercise and Sports, August Krogh Building, University of Copenhagen, 2100 Copenhagen, Denmark; nybo@nexs.ku.dk

**Keywords:** heat stress, thermal stress, hot, heat, occupation, work, assembly line, skin temperature, core temperature, heart rate

## Abstract

Background: The aim of the study was to investigate the effect of a simulated heat-wave on the labour productivity and physiological strain experienced by workers. Methods: Seven males were confined for ten days in controlled ambient conditions. A familiarisation day was followed by three (pre, during, and post-heat-wave) 3-day periods. During each day volunteers participated in a simulated work-shift incorporating two physical activity sessions each followed by a session of assembly line task. Conditions were hot (work: 35.4 °C; rest: 26.3 °C) during, and temperate (work: 25.4 °C; rest: 22.3 °C) pre and post the simulated heat-wave. Physiological, biological, behavioural, and subjective data were collected throughout the study. Results: The simulated heat-wave undermined human capacity for work by increasing the number of mistakes committed, time spent on unplanned breaks, and the physiological strain experienced by the participants. Early adaptations were able to mitigate the observed implications on the second and third days of the heat-wave, as well as impacting positively on the post-heat-wave period. Conclusions: Here, we show for first time that a controlled simulated heat-wave increases workers’ physiological strain and reduces labour productivity on the first day, but it promotes adaptations mitigating the observed implications during the subsequent days.

## 1. Introduction

Appropriate workplace environmental conditions ensure the health and safety of the workforce and the maintenance of productive work. It is therefore not surprising that initiatives to mitigate occupational heat stress have been launched by many organizations, including the World Health Organization [1], the International Labour Organization [2], and the European Commission (HEAT-SHIELD project) [3]. If climate change is not limited, global warming over the next few decades will force humanity to live and work in environments that today are considered extreme. High ambient temperatures are expected to be accompanied by increasing heat-wave frequency, intensity, and duration [4]. Additionally, in contrast to family dwellings in which temperature can be regulated with air-conditioning systems, our current technological advances suggest that it will be probably economically and environmentally inefficient to do the same in large production bays such as the ones found in manufacturing sector [5]. While, it is important to note that even if such heat mitigating measures were adopted in occupational settings, it could lead to a vicious circle of accelerating greenhouse emissions and climate change [5]. Hence, to safeguard workers, this unfortunate cocktail of unavoidable occupational heat stress together with the lack of efficient heat mitigation strategies requires urgent actions on a global scale.

During the last century more than one hundred studies were conducted to investigate the effects of occupational heat stress on workers’ health and productivity [6]. These studies revealed that a single work-shift in the heat quadruples the probability of experiencing occupational heat strain, while one out of three individuals who work under heat stress report productivity losses [6]. Despite this plethora of studies confirming the impact of short-term heat stress on workers’ health and productivity [6,7], no controlled studies have been performed to investigate the cumulative effect of a prolonged heat-wave on the labour productivity and physiological strain experienced by workers.

The likely reason that no attempts have been made to study the effect of heat-waves on worker’s health and productivity in controlled settings is logistics. This is because participants would need to be confined to the temperature-controlled conditions during the work-shift and for the remainder of the day (rest, sleep, etc), simulating the conditions to which the workers would be exposed during such a heat-wave. Therefore, the main gap in our knowledge concerning the effect of heat-waves on heat-health and labour productivity is the cumulative effect of heat-waves on workers. Although the work conducted in previous studies was undoubtedly a step in the right direction, we believe that further study is needed to expand our understanding on this important topic as well as to develop strategies to protect the growing number of people who work in adverse environmental conditions [8]. The present study was designed to address this concern aiming to investigate, for first time, the cumulative effect of a prolonged heat-wave on the labour productivity and physiological strain experienced by people who work in the manufacturing industry.

## 2. Materials and Methods

The experimental protocol was approved by the National Committee for Medical Ethics of the Republic of Slovenia (no. 0120-402/2020/4) in accordance with the Declaration of Helsinki.

### 2.1. Participants

The minimum required sample size for investigating “repeated measures, within factors” was calculated using the results of a previous study [7], which identified that there is a 0.8% increase in labour loss for every degree increase in air temperature (R^2^ = 0.47). Using these data, an effect size (d) of 1.8834 (f = 0.9417) for the association between heat stress and labour loss was computed. Assuming an α of 0.001 and β of 0.99, six participants would provide sufficient power to detect a statistical difference of a similar magnitude (G*Power Version 3.1.9.2) [9]. It is important to note that Eurostat reports that younger (15–39 years) and older (40–59 years) manufacturing workers occupy roughly the same share of jobs in computer and electronics manufacturing industry. However, when a job involves computer programming or other computer-related activities, such as the activities that are required by modern machineries, the industry is dominated by predominantly younger workers (available from: www.appsso.eurostat.ec.eu/nui/submitViewTableAction.do; accessed on 3 March 2021). The same statistics report that male workers occupy 95% more jobs in computer and electronics manufacturing industry, as well as 247% more jobs involving computer programming or other computer-related activities. Hence, the study involves monitoring seven healthy young male students who study in disciplines related to computer science and were not professionally engaged in any sport (age: 21.5 ± 1.2 years (19 to 23 years), weight: 81.5 ± 14.5 kg (69.4 to 115.2 kg), height: 180 ± 5.6 cm (172.5 to 188.5 cm), body surface area: 2.0 ± 0.2 m^2^ (1.9 to 2.4 m^2^), body mass index: 25.1 ± 4.0 kg/m^2^ (20.5 to 33.7 kg/m^2^), fat mass: 22.5 ± 7.9% (15.1 to 40.0%)) during a ten-day confinement experiment. Written informed consent was obtained from all volunteers after detailed explanation of all the procedures involved.

### 2.2. Procedures

Acclimatisation status plays a crucial role in the heat strain experienced by someone. Therefore, to ensure that our participants were not habitually acclimatised prior to the study, the experiments were conducted in autumn. Environmental data including air temperature, relative humidity, and wind speed, two weeks prior to the study were obtained from www.wunderground.com (accessed on 3 March 2021). Wind speed was corrected for height above the ground and air friction coefficient (i.e., large city with tall buildings) using previous methodology [10]. Solar radiation was assumed to be 500 W/m^2^ which is a typical average value for a cloudless day. Liljegren’s method was utilized to compute Wet-Bulb Globe Temperature [11]. Ambient conditions two weeks prior to the study were temperate (19.8 ± 1.8 Wet-Bulb Globe Temperature) indicating that our participants were not exposed to hot conditions, and thus they were not habitually acclimatized.

The study was conducted at the Olympic Sport Centre Planica (Rateče, Slovenia). The participants were confined for ten days to designated areas of the centre in which the ambient temperature and relative humidity were monitored and regulated to simulate the conditions pre, during, and post-heat-wave. Specifically, every day they conducted a simulated work-shift in the laboratory and lived on one floor of the facility for the remainder of the time. Meals were taken in the cafeteria, which was the only area in which the ambient conditions were not controlled and hence always neutral (~23 °C). A total of two hours per day was spent in the cafeteria (breakfast: 40 min, lunch: 40 min, dinner: 40 min). All participants arrived at the Olympic Sport Centre Planica on the same day. The first day (day 0) was dedicated to familiarisation with the experimental protocol. During this day we also obtained baseline measurements of body mass, body stature, and body composition using dual energy X-ray absorption (DXA, Hologic, Discovery W, QDR series; Hologic, Bedford, MA, USA). The participants were also familiarised with the simulated labour duties (i.e., computer tasks and physical activity) they would be conducting during their work-shift in the experimental days of the study. The following nine days were the experimental days, where the participants were exposed to temperate day-time/night-time temperatures on days 1 to 3 (pre-heat-wave) and on days 7 to 9 (post-heat-wave). While, on days 4 to 6 the ambient day-time/night-time temperatures were elevated to simulate the conditions of a heat-wave (Figure 1). Specifically, air temperature was neutral (work = ~25.4 °C and rest = ~22.3 °C) pre/post-heat-wave and hot (work = ~35.4 °C and rest = ~26.3 °C) during the heat-wave (Figure 1). Relative humidity was set to ~45% throughout the experiment. Solar radiation and air velocity were neglectable since our experiments took place in a shaded indoor environment, simulating an indoor manufacturing process. Wet-Bulb globe temperature was neutral (work = ~20.3 °C and rest = ~17.6 °C) pre-/post-heat-wave and hot (work = ~29.0 °C and rest = ~21.1 °C) during the heat-wave.

During the experimental days, all participants underwent a simulated work-shift (duration = 08:40) on a daily basis followed by rest (duration = 07:20) and sleep (duration = 08:00) periods in a controlled environment. A strict time-framed (wake up: 07:00, breakfast: 08:00, work: 08:40–12:00, lunch: 12:00, work: 12:40–18:00, dinner: 18:20, free time: 19:00–23:00 (shower time: 21:40–22:20), and sleep: 23:00) protocol of different psychophysical tasks was followed as described below (Figure 1). To assure that the thermal and physiological strain experienced by our participants was not a product of endogenous factors, we instructed them to restrict physical activity during their free time to essential daily activities. No restrictions were placed on food/water (tap water) consumption, shower temperature, sleeping attire, or any other kind of work or non-work-related behaviour. During all the work-shifts, participants were provided with coveralls (Prevent-Deloza d.o.o., Celje, Slovenia) with an intrinsic clothing insulation (excluding the insulation of the surrounding air layer) equal to 0.61 clo, as determined with the Jozef Stefan Institute thermal manikin (Appendix A).

Due to the inability of ventilation and air conditioning systems that are installed in most industrial settings, the environmental conditions during a heat-wave will more likely affect indoor workplace temperature [12]. For the purpose of this study we simulated the intensity of a well-known heat-wave that occurred in Paris in 2003 [13] and which was responsible for an estimated ~15,000 heat-related deaths. A heat-wave of similar magnitude, but shorter in duration (3 days) was demonstrated to affect the overall equipment effectiveness of the studied factory after its occurrence [12]. Using historical data obtained from www.wunderground.com (accessed on 20 August 2020) we identified that the ambient temperature for the aforementioned Paris heat-wave was increased by 9 °C, compared to the period before and after the heat-wave (average daily temperature: ~30.3 °C vs. ~21.3 °C). Importantly, during that heat-wave, the ambient temperature varied considerably throughout the day, and therefore our experiments were designed to simulate varying levels of heat stress, as shown in Figure 1.

The level of physiological strain experienced by our participants throughout the study was assessed from measurements of body temperature [core temperature (T_core_) and mean skin temperature (T_sk_)], heart rate (HR), subjective ratings (thermal comfort and thermal sensation), and hydration status (urine specific gravity). Specifically, continuous minute by minute T_core_ was measured throughout the study using ingestible telemetric capsules (BodyCap, Caen, France), ingested at the same time (07:00) every morning, immediately after waking up. Similarly, continuous minute by minute skin temperatures were measured throughout the study using wireless thermistors (iButtons type DS1921H, Maxim/Dallas Semiconductor Corp. USA) at four sites, and weighed T_sk_ was determined using Ramanathan’s equation [14]. Beat by beat HR was recorded throughout the study with wireless heart rate monitors (Polar Team2, Polar Electro Oy, Kempele, Finland). Subjective ratings of thermal comfort (1 = comfortable; 5 = extremely uncomfortable) and thermal sensation (0 = cold; 10 = hot) [15] were obtained pre and post each work-related session (see below). Urine samples were collected pre and post each work-shift and were analysed using a handheld refractometer (ATAGO Ltd., Tokyo, Japan). Hydration status was determined on the basis of the urine specific gravity, such that a urine specific gravity of <1.020 was considered a euhydrated state, and a urine specific gravity of ≥1.020 indicated dehydration [16].

Assessing labour productivity requires multi-aspect approaches investigating factors such as (i) the amount of goods produced, (ii) the number of mistakes committed, (iii) the actual time spent doing work, and (iv) the physiological strain experienced by each worker during the work-shift. Therefore, in the current study we assessed labour productivity following a multi-aspect approach, as follows. Our volunteers underwent two 40-min stepping sessions (STEP) on a daily basis at a rate of 12 steps per minute on a 20 cm-stepper (2.8 METs) to simulate the physiological strain experienced by workers during “manual or unskilled labour, light effort”, according to the compendium of physical activities (code: 11475) [17,18]. Each STEP session was followed by a 1-h simulated assembly line task (SALT), which is described in detail below.

SALT software presents participants with images of electronic circuit boards traveling on an assembly line conveyor belt [19] (Figure A2). Participants were required to perform quality control inspections of these electronic circuit boards in order to identify and discard faulty products and repair certain types of defective products. Each circuit board comprised twenty small and two big resistances that the participants were required to measure their resistance using a simulated multimeter. Participants had to decide whether the tested resistance was acceptable (small resistances: 400–600 Ohms; big resistances: 300–700 Ohms) or faulty (i.e., falling outside the determined range). Three possible responses were available for each measured resistance: (i) acceptable, (ii) faulty, but fixable (fix by pressing right click), and (iii) faulty, but not fixable. The latter is discriminated by a high pitch tone instead of a low pitch tone when the resistance is fixable and the text “SHORT” on the screen of the multimeter indicating that there is short circuit in the board, which requires the worker to discard it (by clicking “discard”), before fixing any resistances on that circuit board. Overall, four different types of mistakes were possible: (i) fix a resistance that does not require fixing, (ii) skip fixing a resistance that it needs to be fixed, (iii) fix the same resistance more than one time, and (iv) miss discarding a circuit board that needs to be discarded. The aim of the test was to correctly fix as many circuit boards as possible within each SALT session. The labour efficiency (EFFICIENCY) of our participants was calculated as the average percentage of mistakes committed per board during each SALT session. To simulate the distraction caused by ambient sound during work in assembly lines, all participants conducted SALT in the same workplace at the same time, while using different computers (Figure 2 and Figure A3). Screen brightness and sound volume remained constant throughout the experiments.

Another important aspect of labour productivity assessment is labour effort. This parameter was previously stated to be driven by the primary protective behavioural mechanisms of thermoregulation and was found to play a major role in labour productivity. Specifically, workers have been previously found to spend substantially more time on irregular work breaks when being exposed to increased occupational heat stress [7]. In the current study, irregular work breaks (BREAKS) were considered the amount of time spent going to the toilet, in which the ambient conditions were thermally neutral throughout the experiment. Our hypothesis behind the assessment of toilet visits was that our participants would be more likely to request toilet breaks, which took place in neutral conditions during the simulated heat-wave, compared to the pre and post-heat-wave periods. For this reason, the frequency and the duration of each visit to the toilet was listed for each participant during every work-shift. The participants were not aware of their toilet breaks being recorded.

### 2.3. Data Analysis

Delta (Δ) values were calculated as the differences in the physiological strain (T_core_, T_sk_, and HR) between the average physiological strain experienced by our participants during the pre-heat-wave sleep (sleep time of days 0–2) and the physiological strain during each SALT and STEP sessions. To avoid confusion, it is important to make clear that sleep days 0–2 correspond to the sleep that took place prior to the experimental days 1–3. That is to say, the sleep of day 0 represents the sleep that took place between 23:00 (day 0) and 07:00 (day 1), and so on. Importantly, the simulated heat-wave occurred between the midnight (24:00) of day 3 and the midnight (24:00) of day 6; therefore, the sleep of day 3 was not considered for the calculation of delta values. Using the same approach, Δ values were computed for the physiological strain index (PSI) that was calculated as follows:$$PSI=5\cdot \left(\mathrm{Tcore}\;–\;\mathrm{Tcore}0\right)\;\cdot \;{\left(39.5\;–\;\mathrm{Tcore}0\right)}^{-1}+\;5\cdot \left(\mathrm{HR}\;-\;\mathrm{HR}0\right)\;\cdot \;{\left(180\;-\;\mathrm{HR}0\right)}^{-1}$$
where T_core_0 is the average core temperature during the 8-h sleep for the days prior to the heat-wave (sleep time of days 0–2), and HR0 is the average heart rate during the 8-h sleep for the days prior to the heat-wave (sleep time of days 0–2).

For comparison purposes a reference point reflecting the average physiological strain experienced by our participants during the three days prior to the heat-wave was calculated by averaging the physiological data (T_core_, T_sk_, and HR) collected during the first three experimental days (days 1 to 3) for both the STEP and SALT sessions. Average baseline values for all variables were calculated as follows: baseline = (day 1 + day 2 + day 3)/3. It is important to note that SALT requires a minimum training of five hours for someone to be considered familiarised with the task [19], and therefore only the day prior to the heat-wave (i.e, day 3, while days 0–2 were dedicated to familiarisation) was taken into account in our analyses. Thereafter, repeated measures ANOVA was used to examine possible differences in the EFFICIENCY, BREAKS, and physiological strain experienced by our participants during STEP and SALT sessions between baseline reference point (averages of days 1–3) and each one of the following six days (days 4–9). The same analysis supplemented with Cohen’s effect sizes was conducted to examine possible differences in the urine specific gravity at the end of the work day between the period prior to the heat wave and each one of the following six days (days 4–9). Additionally, the predicted heat strain software [20] was used to project what would be the level of heat strain experienced by a typical worker performing uninterrupted moderate-intensity work (sitting tasks involving moderate effort activities, compendium of physical activities: code 11590) in the same ambient conditions as the ones we simulated during the heat-wave. Pearson’s correlation analyses were conducted to investigate the associations between T_core_ and all the other parameters collected during STEP and SALT sessions. Similarly, correlations were conducted to examine the associations between EFFICIENCY and all the physiological, subjective, and biological variables collected throughout the experiment. Categories were created for the variables of physiological strain index (groups of 0.5 ranging from −0.5 to 5.0), urine specific gravity (groups of 0.004 ranging from 1.000 to 1.032), thermal comfort (groups of 1 ranging from 1 to 5), and thermal sensation (groups of 1 ranging from 4 to 10). Using linear regression models, these categorical variables were used to explain the variance in EFFICIENCY during SALT sessions, as well as the variance in T_core_ during STEP sessions. Similarly, categorical variables of T_core_ (groups of 0.2 °C ranging from 36.2 to 38.4 °C) were used in linear regression models to explain the variance in HR, T_sk_, thermal comfort, and thermal sensation.

Paired sample t-tests were conducted to investigate possible differences in the T_core_ of our participants between the periods of five minutes before and after the shower (i.e., during rest). All statistical analyses were conducted using averages for each SALT and STEP session. The level of significance for all analyses was set at *p* < 0.05. Statistical analyses were conducted using SPSS 27.0 (IBM, Armonk, NY, USA), Origin 2020 (OriginLab Northampton, MA, USA), and Excel spreadsheets (Microsoft Office, Microsoft, Washington, DC, USA). All results are presented as mean ± sd, unless otherwise stated.

## 3. Results

### 3.1. General Findings

We found that the simulated heat-wave had unfavourable impacts on the physiological strain experienced by our participants. Specifically, T_sk_ of our participants was approximately 1.2 °C higher before the start of the work-shifts during the hot days (days 4–6) compared to the neutral ones (days 1–3 and 7–9) (*p* < 0.05). On the other hand, no such significant differences in the T_core_ and HR of our participants were found before the start of the work-shifts. Moreover, heat-wave impaired considerably the EFFICIENCY and physiological strain during SALT and STEP sessions (Figure 3 and Table A1). It is important to note that we predicted that a typical worker performing uninterrupted moderate-intensity work would experience an increase of 1.1 °C T_core_ compared to the only ~0.7 °C we observed in our study.

Interestingly, we found that during the simulated heat-wave, visits to the toilet (neutral ambient temperature) increased by ~19.0% and were 1 min longer in duration (neutral days 3.2 min vs. hot days 4.2 min) (*p* < 0.05)—a finding possibly reflecting behavioural thermoregulation. No significant differences were found in urine specific gravity (i.e., at the end of work day) of our participants between the period prior to the heat-wave and all the experimental days. However, we identified large effect sizes between the period prior to the heat wave (1.0190 ± 0.0081 (-)) and the first (1.0241 ± 0.0049 (-); d = 1.06) and second day (1.0234 ± 0.0040 (-); d = 0.82) of the simulated heat-wave, indicating that a large sample size might yield a statistically significant effect. No significant differences in the T_core_ of our participants were identified between the periods before (37.3 ± 0.3 °C) and after (37.3 ± 0.4 °C) the shower (*p* > 0.05), indicating that shower temperature had no effect on the heat strain experienced by our participants.

### 3.2. Effect of Heat-Wave on Simulated Assembly Line Task

We found significant moderate relationships during SALT sessions between EFFICIENCY and the variables of physiological strain index (r = 0.344, *p* = 0.001), thermal comfort (r = 0.296, *p* = 0.003), and thermal sensation (r = 0.331, *p* = 0.001). On the other hand, although there was a positive trend between EFFICIENCY and urine specific gravity, there was no significant relationship between the two variables (r = 0.192, *p* = 0.060). Moreover, we found that the categorical variables we created were able to meaningfully explain the variance in EFFICIENCY (Figure 4). Specifically, we found that the categorical variables we created for the physiological strain index were able to explain 53% of the variance in EFFICIENCY (F_(1, 9)_ = 8.850_,_
*p* = 0.018), indicating that there is a ~0.48 percentage point increase in the number of mistakes committed in electronic circuit boards for every 1 point increase in the physiological strain index (EFFICIENCY = 2.14588 + 0.47721 * physiological strain index). Although not statistically significant, urine specific gravity categories were able to explain 45% of the variance in EFFICIENCY (F_(1, 7)_ = 4.908_,_
*p* = 0.069), indicating that there is 0.038 percentage points increase in the number of mistakes committed in electronic circuit boards for every 0.001 increase in urine specific gravity points (EFFICIENCY = −35.80742 + 38.38065 * urine specific gravity). Thermal comfort categories were able to explain 89% of the variance in EFFICIENCY (F_(1, 4)_ = 23.419_,_
*p* = 0.017), indicating that there is a ~0.50 percentage point increase in the number of mistakes committed in electronic circuit boards for every 1 point increase in the thermal comfort scale (EFFICIENCY = 2.14951 + 0.50043 * thermal comfort). Similarly, the thermal sensation categories were able to explain 83% of the variance in EFFICIENCY (F_(1, 4)_ = 14.591, *p* = 0.032), indicating that there is a ~0.38 percentage point increase in the number of mistakes committed in electronic circuit boards for every 1 point increase in the thermal sensation scale (EFFICIENCY = 0.76321 + 0.37743 * thermal sensation).

### 3.3. Effect of Heat-Wave on Physical Activity

Moderate relationships during STEP sessions were identified between T_core_ and the variables of T_sk_ (r = 0.373, *p* < 0.001), HR (r = 0.582, *p* < 0.001), thermal comfort (r = 0.301, *p* = 0.003), and thermal sensation (r = 0.281, *p* = 0.005). T_sk_ was moderately related with HR (r = 0.468, *p* < 0.001) and strongly related to the subjective scales of thermal comfort (r = 0.762, *p* < 0.001) and thermal sensation (r = 0.750, *p* < 0.001). HR was related with thermal comfort (r = 0.502, *p* < 0.001) and thermal sensation (r = 0.444, *p* < 0.001). Moreover, we found that the categorical variables we created were able to explain the variance in T_core_ (Figure 5). Specifically, we found that T_core_ categories were able to explain 78% of the variance in HR (F_(1, 9)_ = 32.78066, *p* = 0.00028), indicating that there is ~20 bpm increase in HR for every 1 °C increase in T_core_ (HR = -633.22036 + 19.90182 * T_core_). Similarly, T_core_ categories were able to explain 87% of the variance in T_sk_ (F_(1, 9)_ = 60.6127, *p* = 0.00003), indicating that there is ~1.3 °C increase in T_sk_ for every 1 °C increase in T_core_ (T_sk_ = −13.782 + 1.29864 * T_core_). T_core_ categories were also able to explain 56% of the variance in thermal comfort (F_(1, 9)_ = 11.63484, *p* = 0.00774), indicating that there is ~0.8 point increase in thermal comfort scale for every 1 °C increase in T_core_ (Thermal comfort = -26.19945 + 0.76545 * T_core_). Although not statistically significant, we found that T_core_ categories were able to explain 26% of the variance in thermal sensation (F_(1, 9)_ = 3.24191, *p* = 0.1053), indicating that there is ~0.5 point increase in thermal sensation scale for every 1 °C in T_core_ (Thermal sensation = −11.71527+ 0.49682 * T_core_).

## 4. Discussion

By confining participants to a temperature-controlled environment, we were able to investigate the effect of a simulated heat-wave on the labour productivity and physiological strain experienced by workers in the manufacturing industry. According to the threshold limit values set by the American Conference of Governmental Industrial Hygienist [21], during a heat-wave of a similar magnitude acclimatised workers are expected to spend four to six hours performing moderate work (300 watts (2.9 METs for a standard worker [22])) or six to eight hours performing light work (180 watts (1.7 METs for a standard worker [22])), assuming they work an 8-h shift. However, the absence of acclimatisation which characterizes the workers at the onset of a heat-wave suggests different work-rest cycles. The same guidelines suggest that non-acclimatised individuals should be instructed to spend up to two hours performing moderate work or four to six hours performing light work during an 8-h shift in such conditions [21]. On the other hand, heavy (415 watts (4.0 METs for a standard worker [22])) and very heavy (520 watts (5.0 METs for a standard worker [22])) work tasks should be restricted [21]. Based on this guiding principle, our simulated work-shifts comprised two 40-min sessions of moderate intensity work (STEP; 327 ± 27 watts (determined using workers’ body surface area); 2.8 METs), each followed by 1-h of light intensity work (SALT; 175 ± 14 watts (determined using workers’ body surface area); 1.5 METs), while no heavy or very heavy tasks were involved. Using the predicted heat strain software [20], we computed that if these threshold limit values were not followed, a typical worker performing moderate-intensity work in such heat-wave would experience an increase of 1.1 °C T_core_ compared to the only ~0.7 °C we observed in our study. Despite following the current guidelines which seem to have a protective role in the physiological strain experienced by workers who work in such conditions, we found that the simulated heat-wave increased the number of mistakes committed, the time spent on unplanned breaks, and the physiological strain experienced by workers.

Our findings are in line with previous studies showing that occupational heat stress affects the capacity of workers to meet the cognitive and physical demands of their work. Although the identified increase in the number of mistakes made by our participants during the SALT task was just one percentage point higher during the first day (+1.0%p) and even lower during the second and third days of the heat-wave (+0.5%p); this translates to a 35% increase in the overall number of mistakes committed during the onset of the heat-wave or to a 17% increase throughout the heat-wave compared to the neutral conditions. The economic fallout of this phenomenon [23] might be devastating for small enterprises, with possible spillover effects and irreparable damage to the reputation of the company. The identified heat-induced labour loss involves several physiological mechanisms and requires a multi-aspect analysis. Firstly, a heat-induced increase in the deep body temperature is an important contributing factor able to impair human cognitive performance [24,25] and decision-making [25]. This, in turn, seems to affect the number of mistakes committed during SALT sessions where our participants had to decide whether a randomly generated number falls within a predefined range. Furthermore, hydration state is undoubtedly one of the most important pillars for healthy and productive work. This becomes even more apparent during work under heat stress, where water loss in the form of sweat often exceeds water consumption [26]. Although in the present study there was a non-significant (*p* = 0.060) trend between hydration status and the number of mistakes committed by our participants, we believe that this might be related to the well-known response lag which characterizes the assessment of hydration using urine-specific gravity [27]. It has been demonstrated that cognitive performance is affected considerably by hydration status [28,29], explaining the observed tendency. Another key component of the identified heat-induced increase in the number of mistakes committed by our participants was their thermal comfort/sensation during SALT sessions, supporting previous findings which state that workers report higher labour productivity when their individual thermal satisfaction is greater [30].

Behavioural thermoregulation plays a critical role in mitigating occupational heat stress, and this is especially true in workplaces where labourers are free to self-pace. Our results of increased BREAKS during heat-wave are in line with previous findings indicating that self-pacing in the form of unplanned breaks is driven by the primary protective behavioural mechanism of thermoregulation [7]. An elevation in T_sk_ reflects both the exogenous factors of the surrounding thermal environment as well as the endogenous heat production and cardiovascular responses reflecting the thermal strain experienced by someone. It has been hypothesised that skin temperature provides a feedforward thermal afferent signal to the hypothalamus [31], possibly resulting in a behavioural response to reduce the work being performed, thus decreasing metabolic heat production. The importance of this behavioural mechanism should be considered invaluable to human health, since several cases of exertional heat stroke have been repeatedly reported in settings where highly motivated individuals are involved, and therefore its contribution to thermoregulation is negligible [32].

Heat acclimatisation is a temporary physiological adaptation which begins on the first day of heat exposure, and its full benefits are generally thought to be complete after approximately two weeks [33,34]. This ability of the human species to adapt to different environments describes the main evolutionary advantage which allowed humans to expand their range of habitats. The importance of acclimatisation is demonstrated by the ability of heat-acclimatised individuals performing tasks, which were considered impossible to complete prior to the acclimatisation [33]. Moreover, heat acclimatisation is considered a major determinant improving athletic [35] and cognitive [36] performances, while minimizing the physiological strain experienced by humans during exposure to hot conditions [33]. Some physiological responses (i.e., core temperature, skin temperature, and heart rate) exhibited a transient increase on the first day of the heat-wave, followed by a progressive adaptation over the remaining two days of the heat-wave. This short-term adaptation also mitigated the physiological strain experienced by our participants during the second and third days of the heat-wave. Although the general prescription for heat acclimatisation/acclimation recommends approximately two weeks of intermittent exposure to hot conditions [33], here we found that continuous heat exposure leads to more rapid adaptations which are able to mitigate the physiological strain and maintain cognitive performance. To our knowledge, this is the first time that such adaptive effects of continuous heat exposure on human capacity to perform manual and cognitive work have been observed, and therefore the lack of studies in this area does not allow for further comparisons. However, bearing in mind that 24 h of intermittent heat exposure during a period of twelve days are considered sufficient to promote heat dissipating adaptations [37], we may assume that the 46 h of warm and 26 h of hot conditions experienced by the participants during the simulated heat-wave in the present study initiated some form of physiological adaptations enhancing their capacity for heat dissipation. This is confirmed by our findings which indicate that the heat stress experienced by our participants during the first day of the heat-wave provoked adaptations mitigating the physiological strain they experienced during the subsequent days. Another important aspect of our findings is the identified after-effect of heat acclimatisation/acclimation on the physiological strain experienced by our participants during the period following the heat-wave, which is in line with previous literature suggesting that a number of physiological adaptations following heat acclimation/acclimatisation might have ergogenic potential under cool or temperate conditions [38].

Although in the near future countries are expected to start moving away from the primary sector of agriculture towards the industries of manufacturing and services, the results of this study could not be generalized to all industrial sectors since they do not reflect what happens in agriculture and construction where solar radiation increases the psychophysical strain experienced by workers [7,39]. Moreover, our study includes healthy young males indicating that different population groups should be examined before generalizing the current findings. Although the very narrow subject sample of this study represents most of the people who work in the computer and electronics manufacturing industry and are mainly involved with computer programming or other computer related activities, it is important to consider that there is a large fraction of the industry that is currently occupied by workers of different characteristics. This is especially true since aerobic capacity, sex, age, and other anthropometric parameters play an important role on human capacity for thermoregulation [40]. For instance, males older than 53 and females older than 56 years old are known to be more susceptible to heat stress during work [40]. Hence, further study is needed to elucidate the effect of heat-waves on the physiology and labour productivity of people who work in different manufacturing industries that are dominated by older workers. Additionally, the ongoing climate change is expected to increase heat-wave frequency, intensity, and duration [4] indicating that future studies should also consider examining heat-waves of different magnitudes and durations. Despite its limitations, the novel methodological approach of the present study led to new innovating findings that could be utilized by models [20] and policy makers [41] to promote a healthy and productive work in the face of climate change.

## 5. Conclusions

A simulated heat-wave is accompanied by unfavourable impacts on the human capacity to work, leading to significant labour loss even for individuals that follow existing guidelines for work under heat stress. Early adaptations which occurred in the first day of a heat-wave are able to mitigate labour loss and physiological strain experienced by people during the subsequent days of the heat-wave, as well as being able to elicit aftereffects minimizing the heat strain experienced by workers in the period following the heat-wave. The present study not only contributed to an increased understanding of physiological responses to a heat-wave, but also introduced a novel methodological approach for future studies.

## Figures and Tables

**Figure 1 ijerph-18-03011-f001:**
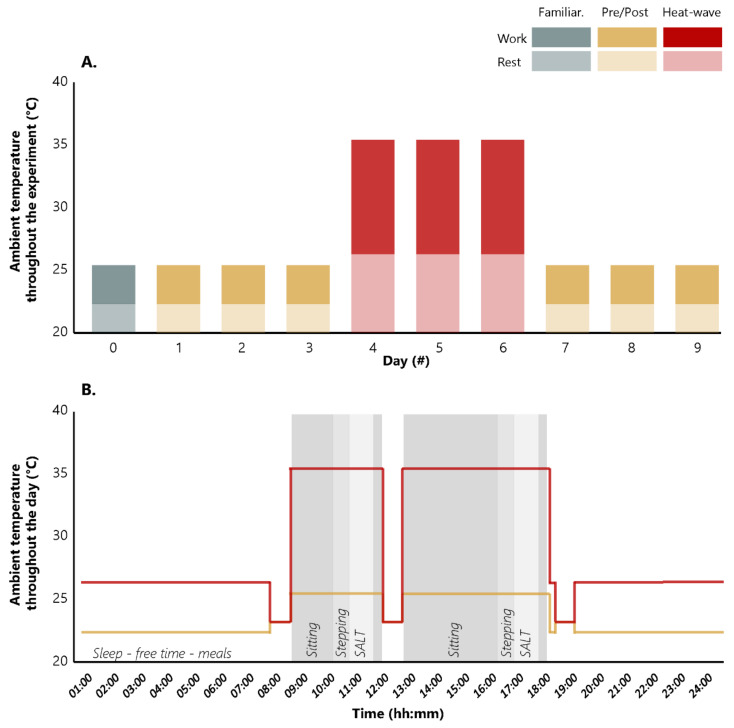
Fluctuation of simulated ambient temperatures throughout the experiment (**A**) and throughout the day (**B**). Turquoise, yellow, and red bars represent familiarisation day, pre/post, and during heat-wave periods, respectively. Dark and light colours correspond to the simulated ambient temperatures during work and rest, respectively. Lines represent the two levels (work and rest) of heat stress during (red) and pre/post (yellow) the simulated heat-wave. Work-shifts were scheduled between 08:40 and 18:00. A strict time-framed (wake up: 07:00, breakfast: 08:00, work: 08:40–12:00, lunch: 12:00, work: 12:40–18:00, dinner: 18:20, free time: 19:00–23:00, and sleep: 23:00) protocol was followed. The time periods spent on passive heat exposure (i.e., sitting in the workplace without having work to perform; duration, 5:20), simulated assembly line tasks (SALT; 2 × 60 min), and stepping sessions (2 × 40 min) are distinguished by different shades of grey. The remaining time was dedicated for meals, free time, and sleep in controlled environmental conditions.

**Figure 2 ijerph-18-03011-f002:**
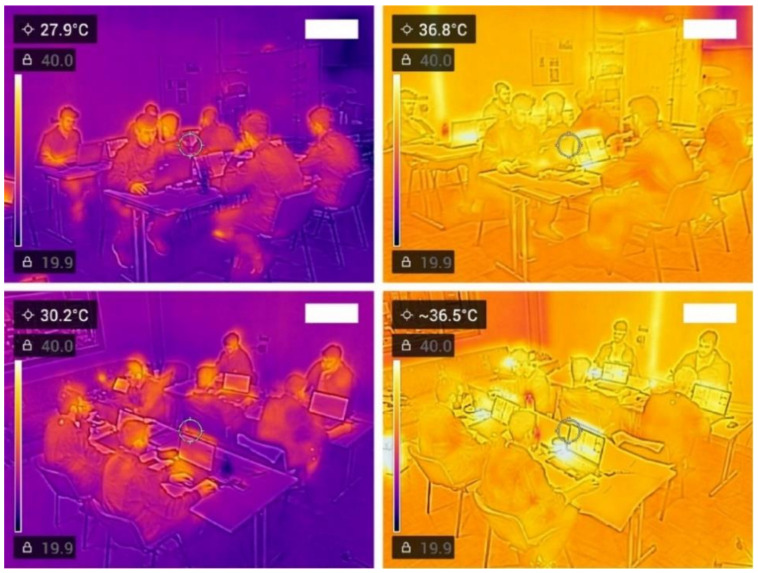
Ambient conditions during the day prior to the heat-wave (**left**) and the first day of the heat-wave (**right**). All pictures were taken using the same thermal camera set to be sensitive within a range of 19.9 and 40.0 °C.

**Figure 3 ijerph-18-03011-f003:**
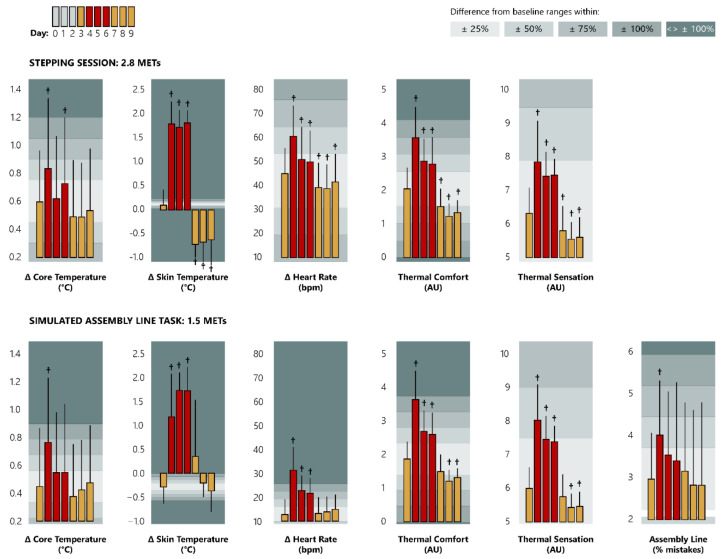
Differences (means ± sd) in thermal strain and labour efficiency during stepping (top graph) and simulated assembly line task (bottom graph) between neutral and hot days. Data are presented as delta (Δ) differences from the pre-heat-wave sleep (sleep time of days 0–2), depicting the increase/decrease in the physiological strain experienced by our participants. Yellow and red bars correspond to the days pre-/post and during the heat-wave, respectively. The first yellow bar (day: 3) represents the average of each variable across the pre-heat-wave period (days 1–3). Grey areas in the background represent the magnitude (%) of difference compared to the pre-heat-wave variables. AU indicates arbitrary units. Cross signs indicate statistically significant differences compared to the pre-heat-wave period, at *p* < 0.05.

**Figure 4 ijerph-18-03011-f004:**
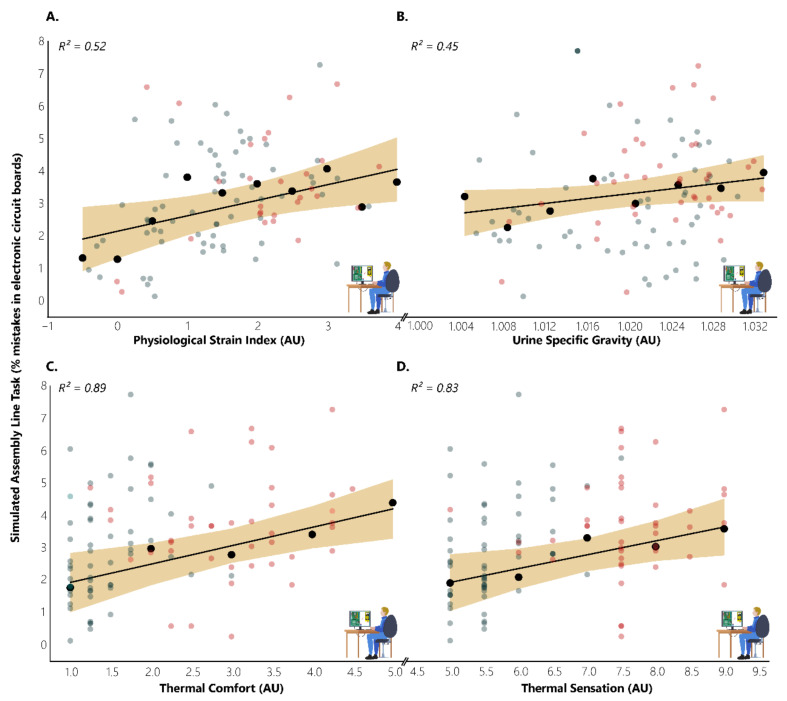
Impacts of physiological strain (**A**), hydration status (**B**), thermal comfort (**C**), and thermal sensation (**D**) on labour productivity, as expressed by physiological strain index, urine specific gravity, and the subjective scales of thermal comfort and thermal sensation, respectively. AU indicates arbitrary units. Red and blue dots represent work during hot (35.4 °C) and neutral (25.4 °C) conditions, respectively. Black dots with their accompanied trendline and 95% CI (shaded area) depict the increase in the number of mistakes committed in electronic circuit boards for every 0.5 points in physiological strain index, 0.004 points in the urine specific gravity scale, and 1 point in the subjective scales of thermal comfort and thermal sensation.

**Figure 5 ijerph-18-03011-f005:**
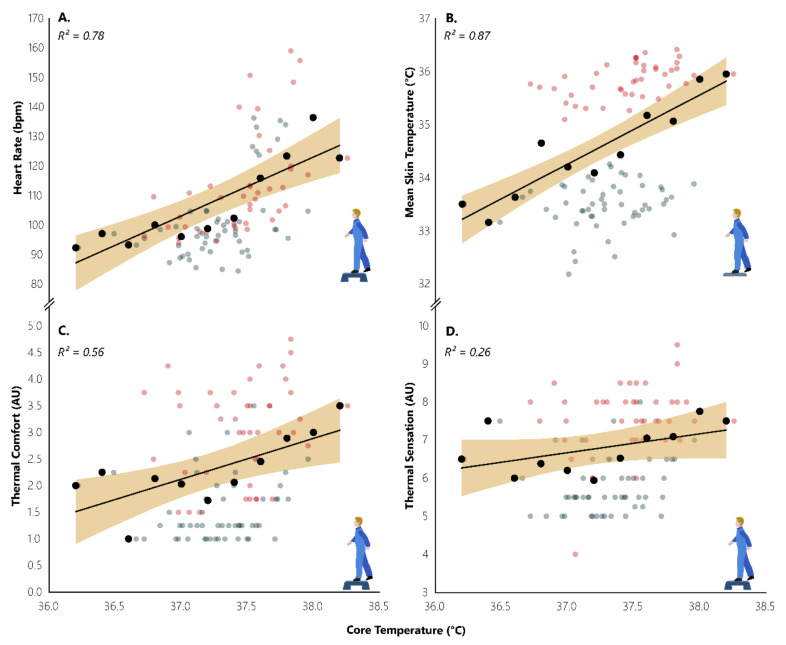
Impacts of core temperature on heart rate (**A**), mean skin temperature (**B**), thermal comfort (**C**), and thermal sensation (**D**). AU indicates arbitrary units. Red and blue dots represent work during hot (35.4 °C) and neutral (25.4 °C) conditions, respectively. Black dots with their accompanied trendline and 95% CI (shaded area) depict the increase in core temperature for every 20-bpm heart rate, 1 °C mean skin temperature, and 1 point in the subjective scales of thermal comfort and thermal sensation.

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
