# Peer review of "Effect of a Simulated Heat Wave on Physiological Strain and Labour Productivity"

_ijerph, 2021, doi:10.3390/ijerph18063011_

Round 1
Reviewer 1 Report
Review of IJERPH 1109116 “Effect of a simulated heat wave on physiological strain and labour productivity” This study looked at work productivity and error rates in seven men subjected to a 3-day simulated heat wave. The study is novel and highly applicable to worker health, safety, and productivity in the changing climate.
The most important issue that needs to be addressed is a better characterization of the subjects in the methods section. Detail is needed on inclusion/exclusion criteria. For example, why only study men? Were they workers, athletes, what was their age range, fitness level, etc? Do their characteristics represent typical age, sex, fitness level of factory workers? As all of these variables can affect the time it might take to acclimatize to heat and heat tolerance, more detail is needed. Similarly, this very narrow subject sample is only given cursory mention as a limitation. As data likely exists on the mean age, health status, and breakdown by gender of industrial workers, more can and should be said about the limitations of this study and possible differences in tolerance, timeline of adaptation, etc. A few minor comments are below.
Abstract, line 11: can remove word controlled and just say 24/7 simulated heat wave.
Abstract, line 17. Rather than stating a 9C increase, provide the range of environmental conditions for work and rest here.
Methods, lines 73-78. These details pertain more to statistical analysis than participant characteristics and should be moved to the end of the methods for data analysis. Instead, this section should detail inclusion and exclusion criteria of the participants.
Figure 1 is extremely confusing. I suggest removing the superimposed Paris heat wave stats and simply using the highs and lows as reference points in the text.
Author Response
Comment 1) Review of IJERPH 1109116 “Effect of a simulated heat wave on physiological strain and labour productivity” This study looked at work productivity and error rates in seven men subjected to a 3-day simulated heat wave. The study is novel and highly applicable to worker health, safety, and productivity in the changing climate.
Thank you for your comments. We made every effort to address your questions and our point-by-point answers appear below. We firmly believe that these changes have improved the overall value of this paper.
Comment 2) The most important issue that needs to be addressed is a better characterization of the subjects in the methods section. Detail is needed on inclusion/exclusion criteria. For example, why only study men? Were they workers, athletes, what was their age range, fitness level, etc? Do their characteristics represent typical age, sex, fitness level of factory workers? As all of these variables can affect the time it might take to acclimatize to heat and heat tolerance, more detail is needed.
Thank you for this comment which helped us to improve the overall quality of our study. The text has been now modified as follows:
[L:74-86] “… difference of a similar magnitude (G*Power Version 3.1.9.2) [9]. It is important to note that, Eurostat reports that younger (15-39 years) and older (40-59 years) manufacturing workers occupy roughly the same share of jobs in computer and electronics manufacturing industry. However, when a job involves computer programming or other computer-related activities, such as the activities that are required by modern machineries, the industry is dominated by predominantly younger workers (available from: www.appsso.eurostat.ec.europa.eu/nui/submitViewTableAction.do). The same statistics report that male workers occupy 95% more jobs in computer and electronics manufacturing industry, as well as 247% more jobs involving computer programming or other computer-related activities. Hence, the study involves monitoring seven healthy young male students who study in disciplines related to computer science and were not professionally engaged in any sport [age: 21.5 ± 1.2 years (19 to 23 years); weight: 81.5 ± 14.5 kg (69.4 to 115.2 kg); height: 180 ± 5.6 cm (172.5 to 188.5 cm); body surface area: 2.0 ± 0.2 m2 (1.9 to 2,4 m2); body mass index: 25.1 ± 4.0 kg/m2 (20.5 to 33.7 kg/m2); fat mass: 22.5 ± 7.9 % (15.1 to 40.0 %)] during a ten-day confinement experiment.”
Comment 3) Similarly, this very narrow subject sample is only given cursory mention as a limitation. As data likely exists on the mean age, health status, and breakdown by gender of industrial workers, more can and should be said about the limitations of this study and possible differences in tolerance, timeline of adaptation, etc.
Thank you for your comment. Our limitation section is now modified as follows:
[L:463-474] “Moreover, our study includes healthy young males indicating that different population groups should be examined before generalizing the current findings. Although the very narrow subject sample of this study represents most of the people who work in the computer and electronics manufacturing industry and are mainly involved with computer programming or other computer related activities, it is important to consider that there is a large fraction of the industry that is currently occupied by workers of different characteristics. This is especially true since aerobic capacity, sex, age, and other anthropometric parameters play an important role on human capacity for thermoregulation [37]. For instance, males older than 53 and females older than 56 years old are known to be more susceptible to heat stress during work [37]. Hence, further study is needed to elucidate the effect of heat-waves on the physiology and labour productivity of people who work in different manufacturing industries that are dominated by older workers. Also, the ongoing climate change…”
A few minor comments are below.
Comment 4) Abstract, line 11: can remove word controlled and just say 24/7 simulated heat wave.
Done.
Comment 5) Abstract, line 17. Rather than stating a 9C increase, provide the range of environmental conditions for work and rest here.
Thank you for this comment which helped us to improve the clarity of our abstract. The text has been modified as follows:
[L:14-17] “During each day volunteers participated in a simulated work-shift incorporating two physical work sessions each followed by a session of assembly line task. Conditions were hot (work: 35.4°C; rest: 26.3°C) during- and temperate (work: 25.4°C; rest: 22.3°C) pre- and post- the simulated heat-wave. Physiological, biological, behavioural, and subjective data were collected throughout the study.”
Comment 6) Methods, lines 73-78. These details pertain more to statistical analysis than participant characteristics and should be moved to the end of the methods for data analysis. Instead, this section should detail inclusion and exclusion criteria of the participants.
While we agree with the reviewer that the literature in previous years tended to present this information in the Methods section, more and more authors nowadays present sample size calculation under “participants” subsection. We agree with the later, as it makes it easier for the reader to make the link between sample size calculation and sample size selection.
Comment 7) Figure 1 is extremely confusing. I suggest removing the superimposed Paris heat wave stats and simply using the highs and lows as reference points in the text.
As suggested, we removed the superimposed Paris heat wave stats from Figure 1. Also, following a comment from Reviewer 2 we modified Figure 1 to depict the simulated ambient conditions throughout the experiment. The legends is now modified as follows:
[L:148-159] Figure 1. Fluctuation of simulated ambient temperatures throughout the experiment (graph A) and throughout the day (graph B). Turquoise, yellow, and red bars represent familiarisation day, pre-/post-, and during- heat-wave- periods, respectively. Dark and light colours correspond to the simulated ambient temperatures during work and rest, respectively. Lines represent the two levels (work and rest) of heat stress during (red) and pre-/post- (yellow) the simulated heat-wave. Work-shifts were scheduled between 08:40 and 18:00. A strict time-framed (wake up: 07:00, breakfast: 08:00, work: 08:40-12:00, lunch: 12:00, work: 12:40-18:00, dinner: 18:20, free time: 19:00-23:00, and sleep: 23:00) protocol was followed. The time periods spent on passive heat exposure (i.e., sitting in the workplace without having work to perform; duration, 5:20), simulated assembly line tasks (SALT; 2 x 60 minutes), and stepping sessions (2 x 40 minutes) are distinguished by different shades of grey. The remaining time was dedicated for meals, free time, and sleep in controlled environmental conditions.

Reviewer 2 Report
The authors have presented a very interesting study evaluating the effect of a heatwave on occupational stress. Please see comments below.
Major concerns –
- The authors need to be clearer in the ambient temperature that their participants were exposed to. The way it is written, people were exposed to a constant temperature depending on whether it was a heat wave day or not. However when looking at figure one, it is clear that this is not the case.
- From the data and statistical analysis, it is unclear how the authors calculated baseline values and over which day’s baseline values were obtained.
- The manuscript is well written however at times it is very hard to follow. The use of subheadings would improve the flow of the article. Provide sub headings for the Step test, the salt test and during the results section.
Minor concerns
Introduction
- The rationale behind the study is sound, however the authors should state their hypothesis based on the background information. Perhaps the authors should also state what type of workers they are focusing on, industrial or trade (i.e. carpenters).
- What is the difference between heat stress caused by a hot environment and heat stress caused by a heat wave?
Methods
- The authors have justified their sample size, however could the authors justify why they chose people of this age group? Most individuals who work in warehouse settings/ who perform laborious work are an average age of 40.
- Have the authors taken note of the normal daily activities of the participants? Were these participants physically active, what did they normally do for work? Is it possible that these participants are partially acclimated through exercise and work?
- What time of year was the data collection?
- Was the ambient temperature manually controlled to oscillate across the day? If not, what temperature was the room set to? Looking at figure 1, it looks like the workers only spent under four hrs in the heatwave environment and the rest of the time was spent working in a ~30 degree Celsius room.
- The authors should consider providing an additional systematic timeline of the study. For example day 0 – anthropometric data, day 1-3 pre heatwave (then provide the temperature), etc. this information is already provided within the text, but it is hard to understand at times, particularly when accompanied by figure one, where the temperature of the room seems to change considerably across the day.
- Figure one is slightly confusing to interpret with the dashed lines. It seems unclear why the authors have marked work on the Y and X axis.
- Was sleeping attire standardized?
- P3, L118. Perhaps you could replace the term physical work with physical activity, to separate the ideas of working and exercising.
- P 3, L119 – Tap water, not tab water
- How often did participants swallow a new temperature pill? And how was this monitored? Typically, a telemetric pill is ingested ~6 h prior to a study to avoid any influence of temperature changes along the gastrointestinal tract. How did the authors account for this
Data analysis
- How often were the authors recording core temperature of the participants? Particularly when sleeping, did the authors have a constant record of core temperature?
- Was there any control for shower temperature? To avoid the participants have very cold showers?
- P 6, L217. Here you write the pre heat wave days are 0-2, where previously you write the pre heat wave days are 1-3 (with day 0 being used to collect anthropometric data). Please be consistent with this.
- Did the authors combine and average all skin and core temperature and heart rate data from days 1-3 and use this as the baseline data? It would be important to state the sampling rate of each measure and how the data were average across time.
- P 6, L233 – here the authors use days 1-3 instead of days 0-2.
- sampling rates should be provided for all data
Results
- The results section should be divided into sub categories to make it easier to follow. As it stands now, there is a lot of text.
- Could the authors please provide data for core and skin temperature and the changes across each day? Alongside physiological strain, it would be worth providing information in a table as to what the absolute and change in core and skin temperature were for each day.
- What was the ambient temperature when the participants were sleeping during the heatwave day?
- Figure 3 – Does the first yellow bar of the graph represent the average of each variable across a 3 day period (i.e. day 1-3). If so, does that value represent the total change from day 1 to 3? And therefore, are the values for the following red and yellow bars the difference in the change of each variable? It would help to understand the figure if the authors placed data points on the x-axis.
- Was USG different at the end of a work day during the heatwave days compared to neutral days? And were the participants consuming enough water to offset any dehydration? did fluid consumption change on heatwave days compared to neutral days?
- The analysis for the step task and the variable determinants of core temperature seem unnecessary. In fact, it would seem more logical to report the information the other way around. For example, thermal sensation and comfort are more likely to change as a result of changes in core temperature, but it is physiologically impossible for core temperature to change based on thermal sensation and comfort. Perhaps this is what the authors are trying to write, however what would influence core temperature are the environmental conditions, work, and the ability to produce and evaporate sweat, not perceptual measures.
- What were the total hours that participants were exposed to the heatwave at work and during rest? i understand that they performed ~8 h of work, but from figure 1, it seems as though not all of this time was with exposure to ambient temperature of ~35 degrees.
Discussion and conclusion
- P 9, L 329. The term 24/7 implies 24 hours, 7 days a week, which was not the case in this study. The authors exposed people to a heat wave for 3 days, and it is my understanding that this was not for a full 24 h.
- P 9, L335. The authors should change their measure in watts to mets to have a comparison to the amount of work that was performed during this study.
- P 10, L 349, these are results that you have not presented in the results section.
- For the conclusion, I raise the same concern over the use of ‘24/7’.
Author Response
REVIEWER #2
The authors have presented a very interesting study evaluating the effect of a heatwave on occupational stress. Please see comments below.
Thank you for your kind words. We made every effort to address your questions and our point-by-point answers appear below. We firmly believe that these changes have improved the overall value of this paper.
Major concerns –
Comment 1) The authors need to be clearer in the ambient temperature that their participants were exposed to. The way it is written, people were exposed to a constant temperature depending on whether it was a heat wave day or not. However when looking at figure one, it is clear that this is not the case.
Thank you for this comment which helped us to improve the overall quality of our study. Several changes, including modifying Figure 1, have been implemented to improve clarity.
[L:14-17] “During each day volunteers participated in a simulated work-shift incorporating two physical work sessions each followed by a session of assembly line task. Conditions were hot (work: 35.4°C; rest: 26.3°C) during- and temperate (work: 25.4°C; rest: 22.3°C) pre- and post- the simulated heat-wave. Physiological, biological, behavioural, and subjective data were collected throughout the study.”
[L:148-151] “Figure 1. Ambient temperatures throughout the experiment (graph A) and throughout the day (graph B). Turquoise, yellow, and red bars represent familiarisation day, pre-/post-, and during- heat-wave- periods, respectively. Dark and light colours correspond to the simulated ambient temperatures during work and rest, respectively.…”
Comment 2) From the data and statistical analysis, it is unclear how the authors calculated baseline values and over which day’s baseline values were obtained.
Thank you for this comment which helped us to improve clarity. Our data analysis sections has been modified as follows:
[L:226-232] “…participants during the pre-heat-wave sleep (sleep time of days: 0-2) and the physiological strain during each SALT and STEP sessions. To avoid confusion, it is important to make clear that sleep days 0-2 correspond to the sleep that took place prior to the experimental days 1-3. That is to say, the sleep of day 0 represents the sleep that took place between 23:00 (day 0) and 07:00 (day 1), and so on. Importantly, the simulated heat-wave occurred between the midnight (24:00) of day 3 and the midnight (24:00) of day 6, therefore the sleep of day 3 was not considered for the calculation of delta values.”
[L:235-237] “…to the heat-wave (sleep time of days: 0-2) and HR0 is the average heart rate during the 8-hour sleep for the days prior to the heat-wave (sleep time of days: 0-2).”
[L:239-243] “For comparison purposes a reference point reflecting the average physiological strain experienced by our participants during the three days prior to the heat-wave was calculated by averaging the physiological data (Tcore, Tsk, and HR) collected during the first three experimental days (days: 1 to 3) for both the STEP and SALT sessions. Average baseline values for all variables were calculated as follows: baseline = (day 1 + day 2 + day 3) / 3.”
Comment 3) The manuscript is well written however at times it is very hard to follow. The use of subheadings would improve the flow of the article. Provide sub headings for the Step test, the salt test and during the results section.
As suggested, our Results section has been now divided into three sub-sections as follows:
[L:277] “3.1. General findings”
[L:310] “3.2. Effect of heat-wave on simulated assembly line task”
[L:342] “3.3. Effect of heat-wave on physical activity”
Minor concerns
Introduction
Comment 4) The rationale behind the study is sound, however the authors should state their hypothesis based on the background information. Perhaps the authors should also state what type of workers they are focusing on, industrial or trade (i.e. carpenters).
Our text is now modified as follows:
[L:61-63] “…the cumulative effect of a prolonged heat-wave on the labour productivity and physiological strain experienced by people who work in the manufacturing industry.
Comment 5) What is the difference between heat stress caused by a hot environment and heat stress caused by a heat wave?
To improve clarity our text is now modified as follows:
[L:48-51] “Despite this plethora of studies confirming the impact of short-term heat stress on workers’ health and productivity [6, 7], no controlled studies have been performed to investigate the cumulative effect of a prolonged heat-wave on the labour productivity and physiological strain experienced by workers.”
Methods
Comment 6) The authors have justified their sample size, however could the authors justify why they chose people of this age group? Most individuals who work in warehouse settings/ who perform laborious work are an average age of 40.
Thank you for this comment which helped us to improve the overall quality of our study. The text has been now modified as follows:
[L:74-82] “… difference of a similar magnitude (G*Power Version 3.1.9.2) [9]. It is important to note that, Eurostat reports that younger (15-39 years) and older (40-59 years) manufacturing workers occupy roughly the same share of jobs in computer and electronics manufacturing industry. However, when a job involves computer programming or other computer-related activities, such as the activities that are required by modern machineries, the industry is dominated by predominantly younger workers (available from: www.appsso.eurostat.ec.europa.eu/nui/submitViewTableAction.do). The same statistics report that male workers occupy 95% more jobs in computer and electronics manufacturing industry, as well as 247% more jobs involving computer programming or other computer-related activities.”
Comment 7) Have the authors taken note of the normal daily activities of the participants? Were these participants physically active, what did they normally do for work? Is it possible that these participants are partially acclimated through exercise and work?
The text is now modified as follows:
[L:82-86] “…the study involves monitoring seven healthy young male students who study in disciplines related to computer science and were not professionally engaged in any sport [age: 21.5 ± 1.2 years (19 to 23 years); weight: 81.5 ± 14.5 kg (69.4 to 115.2 kg); height: 180 ± 5.6 cm (172.5 to 188.5 cm); body surface area: 2.0 ± 0.2 m2 (1.9 to 2,4 m2); body mass index: 25.1 ± 4.0 kg/m2 (20.5 to 33.7 kg/m2); fat mass: 22.5 ± 7.9 % (15.1 to 40.0 %)] during a ten-day confinement experiment.”
Comment 8) What time of year was the data collection?
Thank you for this comment which improved considerably the quality of our manuscript. The text is now modified as follows:
[L:90-99] “Acclimatisation status plays a crucial role in the heat strain experienced by someone. Therefore, to ensure that our participants were not habitually acclimatised prior to the study, the experiments were conducted in autumn. Environmental data including air temperature, relative humidity and wind speed, two weeks prior to the study were obtained from www.wunderground.com. Wind speed was corrected for height above the ground and air friction coefficient (i.e., large city with tall buildings) using previous methodology [10]. Solar radiation was assumed to be 500 W/m2 which is a typical average value for a cloudless day. Liljegren’s method was utilized to compute Wet-Bulb Globe Temperature [11]. Ambient conditions two weeks prior to the study were temperate (19.8 ± 1.8 Wet-Bulb Globe Temperature) indicating that our participants were not exposed to hot conditions and thus they were not habitually acclimatized.”
Comment 9) Was the ambient temperature manually controlled to oscillate across the day? If not, what temperature was the room set to? Looking at figure 1, it looks like the workers only spent under four hrs in the heatwave environment and the rest of the time was spent working in a ~30 degree Celsius room.
To further improve clarity Figure 1 is now modified to show the fluctuation of ambient temperature throughout the day. Moreover, the following information is now added in the caption of Figure 1.
[L:152-155] “Work-shifts were scheduled between 08:40 and 18:00. A strict time-framed (wake up: 07:00, breakfast: 08:00, work: 08:40-12:00, lunch: 12:00, work: 12:40-18:00, dinner: 18:20, free time: 19:00-23:00, and sleep: 23:00) protocol was followed.”
Comment 10) The authors should consider providing an additional systematic timeline of the study. For example day 0 – anthropometric data, day 1-3 pre heatwave (then provide the temperature), etc. this information is already provided within the text, but it is hard to understand at times, particularly when accompanied by figure one, where the temperature of the room seems to change considerably across the day.
To further improve clarity Figure 1 is now modified to show the fluctuations of ambient temperature throughout the experiment. Moreover, the following information is now added in the caption of Figure 1.
[L:148-151] “Figure 1. Fluctuation of simulated ambient temperatures throughout the experiment (graph A) and throughout the day (graph B). Turquoise, yellow, and red bars represent familiarisation day, pre-/post-, and during- heat-wave- periods, respectively. Dark and light colours correspond to the simulated ambient temperatures during work and rest, respectively.”
Comment 11) Figure one is slightly confusing to interpret with the dashed lines. It seems unclear why the authors have marked work on the Y and X axis.
Following your comments Figure 1 is now modified.
Comment 12) Was sleeping attire standardized?
Since the aim of our study was to simulate the physiological strain experienced by our participants during the occurrence of a heat-wave, we did not place any restrictions on behavioural thermoregulation, including controlling sleeping attire. Our text in now modified as follows:
[L:130-131] “No restrictions were placed on food/water (tap water) consumption, shower temperature, sleeping attire, or any other kind of work- or non-work-related behaviour.”
Comment 13) P3, L118. Perhaps you could replace the term physical work with physical activity, to separate the ideas of working and exercising.
The term “physical work” has been replaced with “physical activity” throughout the manuscript.
Comment 14) P 3, L119 – Tap water, not tab water
Done, thank you for noting this typo.
Comment 15) How often did participants swallow a new temperature pill? And how was this monitored? Typically, a telemetric pill is ingested ~6 h prior to a study to avoid any influence of temperature changes along the gastrointestinal tract. How did the authors account for this
We respectfully disagree with the Reviewer. A recent study published by a Canadian group shows that time following ingestion does not influence the validity of telemetry pill measurements of core temperature (DOI: 10.1080/23328940.2020.1801119). To further improve clarity, our text is now modified as follows:
[L:165-166] “… using ingestible telemetric capsules (BodyCap, Caen, France), ingested at the same time (07:00) every morning, immediately after waking up.”
Data analysis
Comment 16) How often were the authors recording core temperature of the participants? Particularly when sleeping, did the authors have a constant record of core temperature?
Thank you for this comment. After a full (24-hour) day, some nocturnal Tcore data points were lost due to the need of our participants for defecation. However, those nocturnal sleeping data are not part of this study. To improve clarity, the text is now modified as follows:
[L:164-166] “Specifically, continuous minute by minute Tcore was measured throughout the study using ingestible telemetric capsules (BodyCap, Caen, France), ingested at the same time (07:00) every morning, immediately after waking up.”
Comment 17) Was there any control for shower temperature? To avoid the participants have very cold showers?
To improve clarity our manuscript has been modified as follows:
[L:125-127] “A strict time-framed [wake up: 07:00, breakfast: 08:00, work: 08:40-12:00, lunch: 12:00, work: 12:40-18:00, dinner: 18:20, free time: 19:00-23:00 (shower time: 21:40-22:20), and sleep: 23:00] protocol of different…”
[L:130-131] “No restrictions were placed on food/water (tap water) consumption, shower temperature, sleeping attire, or any other kind of work- or non-work-related behaviour.”
[L:268-269] “Paired sample t-tests were conducted to investigate possible differences in the Tcore of our participants between the periods of five minutes before and after the shower (i.e., during rest).”
[L:294-297] “significant differences in the Tcore of our participants were identified between the periods before (37.3±0.3 °C) and after (37.3±0.4 °C) the shower (p > 0.05), indicating that shower temperature had no effect on the heat strain experienced by our participants.”
Comment 18) P 6, L217. Here you write the pre heat wave days are 0-2, where previously you write the pre heat wave days are 1-3 (with day 0 being used to collect anthropometric data). Please be consistent with this.
Thank you for this comment which helped us to improve clarity. Our data analysis sections has been modified as follows:
[L:226-232] “…participants during the pre-heat-wave sleep (sleep time of days: 0-2) and the physiological strain during each SALT and STEP sessions. To avoid confusion, it is important to make clear that sleep days 0-2 correspond to the sleep that took place prior to the experimental days 1-3. That is to say, the sleep of day 0 represents the sleep that took place between 23:00 (day 0) and 07:00 (day 1), and so on. Importantly, the simulated heat-wave occurred between the midnight (24:00) of day 3 and the midnight (24:00) of day 6, therefore the sleep of day 3 was not considered for the calculation of delta values.”
[L:235-237] “…to the heat-wave (sleep time of days: 0-2) and HR0 is the average heart rate during the 8-hour sleep for the days prior to the heat-wave (sleep time of days: 0-2).”
Comment 19) Did the authors combine and average all skin and core temperature and heart rate data from days 1-3 and use this as the baseline data? It would be important to state the sampling rate of each measure and how the data were average across time.
To improve clarity our manuscript has been modified as follows:
[L:164-170] “Specifically, continuous minute by minute Tcore was measured throughout the study using ingestible telemetric capsules (BodyCap, Caen, France), ingested at the same time (07:00) every morning, immediately after waking up. Similarly, continuous minute by minute skin temperatures were measured throughout the study using wireless thermistors (iButtons type DS1921H, Maxim/Dallas Semiconductor Corp. USA) at four sites and weighed Tsk was determined using Ramanathan’s equation [14]. Beat by beat HR was recorded throughout the study with wireless heart rate monitors…”
[L:270] “All statistical analyses were conducted using averages for each SALT and STEP session. ”
[L:239-243] “For comparison purposes a reference point reflecting the average physiological strain experienced by our participants during the three days prior to the heat-wave was calculated by averaging the physiological data (Tcore, Tsk, and HR) collected during the first three experimental days (days: 1 to 3) for both the STEP and SALT sessions. Average baseline values for all variables were calculated as follows: baseline = (day 1 + day 2 + day 3) / 3.”
Comment 20) P 6, L233 – here the authors use days 1-3 instead of days 0-2.
This comment has been answered above (comment 18).
Comment 21) sampling rates should be provided for all data
Thank you for this comment which helped us to improve clarity. The text has been modified as follows:
[L:164-170] “Specifically, continuous minute by minute Tcore was measured throughout the study using ingestible telemetric capsules (BodyCap, Caen, France), ingested at the same time (07:00) every morning, immediately after waking up. Similarly, continuous minute by minute skin temperatures were measured throughout the study using wireless thermistors (iButtons type DS1921H, Maxim/Dallas Semiconductor Corp. USA) at four sites and weighed Tsk was determined using Ramanathan’s equation [14]. Beat by beat HR was recorded throughout the study with wireless heart rate monitors…”
Results
Comment 21) The results section should be divided into sub categories to make it easier to follow. As it stands now, there is a lot of text.
Our Results section has been now divided into three sub-sections as follows:
[L:277] “3.1. General findings”
[L:310] “3.2. Effect of heat-wave on simulated assembly line task”
[L:342] “3.3. Effect of heat-wave on physical activity”
Comment 22) Could the authors please provide data for core and skin temperature and the changes across each day? Alongside physiological strain, it would be worth providing information in a table as to what the absolute and change in core and skin temperature were for each day.
We agree with the reviewer. Average, SD, minimum, and maximum values are now presented in Table C1. Due to the large size of the Table, it was moved to Appendix C and it is referenced as follows:
[L:282-284] “Moreover, heat-wave impaired considerably the EFFICIENCY and physiological strain during SALT and STEP sessions (Figure 3 and Table C1).”
Comment 23) What was the ambient temperature when the participants were sleeping during the heatwave day?
To improve clarity Figure 1 is now modified to show the fluctuation of ambient temperature throughout the day, including sleep time (23:00 – 07:00).
Comment 24) Figure 3 – Does the first yellow bar of the graph represent the average of each variable across a 3 day period (i.e. day 1-3). If so, does that value represent the total change from day 1 to 3? And therefore, are the values for the following red and yellow bars the difference in the change of each variable? It would help to understand the figure if the authors placed data points on the x-axis.
Thank you for your comment which helped us to improve clarity. We redrew Figure 3 and the accompanied legend is now modified as follows:
[L:300-308] “Figure 3. Differences (means ± sd) in thermal strain and labour efficiency during stepping (top graph) and simulated assembly line task (bottom graph) between neutral and hot days. Data are presented as delta (Δ) differences from the pre-heat-wave sleep (sleep time of days: 0-2), depicting the increase/decrease of the physiological strain experienced by our participants. Yellow and red bars correspond to the days pre-/post- and during- the heat-wave, respectively. The first yellow bar (day: 3) represents the average of each variable across the pre heat-wave period (days: 1-3). Grey areas in the background represent the magnitude (%) of difference compared to the pre- heat-wave variables. AU indicates arbitrary units. Cross signs indicate statistically significant differences compared to the pre-heat-wave period, at p < 0.05.”
Comment 25) Was USG different at the end of a work day during the heatwave days compared to neutral days? And were the participants consuming enough water to offset any dehydration? did fluid consumption change on heatwave days compared to neutral days?
Additional analyses were conducted and are presented in the manuscript as follows:
[L:249-252] “The same analysis supplemented with Cohen’s effect sizes was conducted to examine possible differences in the urine specific gravity at the end of the work day between the period prior to the heat wave and each one of the following six days (days: 4-9).”
[L:289-294] “No significant differences were found in urine specific gravity (i.e., at the end of work day) between the period prior to the heat-wave and all the experimental days. However, we identified large effect sizes between the period prior to the heat wave (1.0190 ± 0.0081 [-]) and the first (1.0241 ± 0.0049 [-]; d = 1.06) and second day (1.0234 ± 0.0040 [-]; d = 0.82) of the simulated heat-wave, indicating that a large sample size might yield a statistically significant effect.”
Comment 26) The analysis for the step task and the variable determinants of core temperature seem unnecessary. In fact, it would seem more logical to report the information the other way around. For example, thermal sensation and comfort are more likely to change as a result of changes in core temperature, but it is physiologically impossible for core temperature to change based on thermal sensation and comfort. Perhaps this is what the authors are trying to write, however what would influence core temperature are the environmental conditions, work, and the ability to produce and evaporate sweat, not perceptual measures.
We totally agree with the reviewer. For this reason, this information is now presented the other way around. Specifically, we redrew Figure 5 and reconducted our analyses as follows:
[L:265-267] “during STEP sessions. Similarly, categorical variables of Tcore (groups of 0.2 °C ranging from 36.2 to 38.4°C) were used in linear regression models to explain the variance in HR, Tsk, thermal comfort, and thermal sensation.”
[L:348-359] “Moreover, we found that the categorical variables we created were able to explain the variance in Tcore (Figure 5). Specifically, we found that Tcore categories were able to explain 78 % of the variance in HR (F(1, 9) = 32.78066, p = 0.00028), indicating that there is ~20 bpm increase in HR for every 1 °C increase in Tcore (HR = -633.22036 + 19.90182 * Tcore). Similarly, Tcore categories were able to explain 87 % of the variance in Tsk (F(1, 9) = 60.6127, p = 0.00003), indicating that there is ~1.3 °C increase in Tsk for every 1 °C increase in Tcore (Tsk = -13.782 + 1.29864 * Tcore). Tcore categories were also able to explain 56 % of the variance in thermal comfort (F(1, 9) = 11.63484, p = 0.00774), indicating that there is ~0.8 point increase in thermal comfort scale for every 1 °C increase in Tcore (Thermal comfort = -26.19945 + 0.76545 * Tcore). Although not statistically significant, we found that Tcore categories were able to explain 26 % of the variance in thermal sensation (F(1, 9) = 3.24191, p = 0.1053), indicating that there is ~0.5 point increase in thermal sensation scale for every 1 °C in Tcore (Thermal sensation = -11.71527+ 0.49682 * Tcore).”
[L:362] “Figure 5. Impacts of core temperature on heart rate (graph A), mean skin temperature (graph B)…”
Comment 27) What were the total hours that participants were exposed to the heatwave at work and during rest? i understand that they performed ~8 h of work, but from figure 1, it seems as though not all of this time was with exposure to ambient temperature of ~35 degrees.
To further improve clarity Figure 1 is now modified to show the fluctuation of ambient temperature throughout the day. Moreover, the following information is now added in the caption of Figure 1:
[L:XX-XX] “Work-shifts were scheduled between 08:40 and 18:00. A strict time-framed (wake up: 07:00, breakfast: 08:00, work: 08:40-12:00, lunch: 12:00, work: 12:40-18:00, dinner: 18:20, free time: 19:00-23:00, and sleep: 23:00) protocol was followed.”
Discussion and conclusion
Comment 28) P 9, L 329. The term 24/7 implies 24 hours, 7 days a week, which was not the case in this study. The authors exposed people to a heat wave for 3 days, and it is my understanding that this was not for a full 24 h.
As suggested the term 24/7 has been removed throughout the text.
Comment 29) P 9, L335. The authors should change their measure in watts to mets to have a comparison to the amount of work that was performed during this study.
Thank you for the suggestion. The text has been modified as follows:
[L:374-376] “…performing moderate work [300 watts (2.9 METs for a standard worker [21])] or six to eight hours performing light work [180 watts (1.7 METs for a standard worker [21])], assuming…”
[L:380-382] “On the other hand, heavy [415 watts (4.0 METs for a standard worker [21])] and very heavy [520 watts (5.0 METs for a standard worker [21])] work tasks should…”
[L:383-385] “…intensity work [STEP; 327±27 watts (determined using workers’ body surface area); 2.8 METs], each followed by 1-hour of light intensity work [SALT; 175±14 watts (determined using workers’ body surface area); 1.5 METs]…”
Comment 30) P 10, L 349, these are results that you have not presented in the results section.
To improve clarity our text has been modified as follows:
[L:252-256] “Additionally, the predicted heat strain software [20] was used to project what would be the level of heat strain experienced by a typical worker performing uninterrupted moderate-intensity work (sitting tasks involving moderate effort activities; compendium of physical activities: code 11590) in the same ambient conditions as the ones we simulated during the heat-wave.”
[L:284-286] “It is important to note that, we predicted that a typical worker performing uninterrupted moderate-intensity work would experience an increase of 1.1°C Tcore compared to the only ~0.7°C we observed in our study.”
Comment 31) For the conclusion, I raise the same concern over the use of ‘24/7’.
Following your previous comment (comment 28), the term 24/7 has been removed throughout the text.

Round 2
Reviewer 2 Report
They authors have adequatley answered all of my comments. Well done for presenting a wonderful study.
This manuscript is a resubmission of an earlier submission. The following is a list of the peer review reports and author responses from that submission.